# Mamba-HMIL: Hierarchical Multiple Instance Learning via State Space Model for Whole Slide Image Diagnosis

## Abstract

Multiple instance learning (MIL) has been widely employed for gigapixel whole slide image (WSI) diagnosis. Existing MIL methods, however, are found wanting to align with the clinical practice of pathologists, who typically scrutinize WSIs at varied scales and compare the local regions in a global perspective. Given that WSIs usually boast immense dimensions peppered with large regions not pertinent to diagnosis, we propose a novel hierarchical multiple instance learning method based on the state space model (SSM) , called **Mamba-HMIL**, for WSI classification. **Mamba-HMIL** consists of three primary modules to enhance the performance of MIL. First, the hierarchical feature extractor harvests features across diverse scales. Second, for capturing the correlation among patches, the state space model demonstrates robust modeling capabilities. A Mixture of Experts (MoE) module is for stable SSM training. Third, the adaptive selection model strives to reduce redundancies by focusing on disease-positive regions. We evaluate **Mamba-HMIL** on two WSI subtype datasets (TCGA-NSCLC and TCGA-RCC) and two WSI survival datasets (TCGA-BRCA and TCGA-BLCA). Our results suggest that **Mamba-HMIL** outperforms existing MIL methods on both WSI tasks. Our code will be made publicly available.

## 1 Introduction

Pathological image analysis serves as the gold standard for cancer diagnosis Kumar et al. (2014). Rapid advancements in scanning technologies Farahani et al. (2015) have digitized pathological scans into whole slide images (WSIs) of up to $100,000 \times 100,000$ pixels. Analyzing these WSIs can be a labor-intensive and time-consuming task that demands considerable expertise and concentration from pathologists Evered & Dudding (2011). Recent studies indicate that computer-aided methods could alleviate these demands Tizhoosh & Pantanowitz (2018); Bera et al. (2019); Niazi et al. (2019); Colling et al. (2019); Jiang et al. (2020). However, due to the immensity of WSIs, computer-aided analysis needs huge computational resources,posing a considerable challenge Evered & Dudding (2011). To address this, researchers have cropped each WSI into a large number of patches, which can be treated as a bag of instances. Thus, cancer diagnosis using WSIs has been formulated into a multiple instance learning (MIL) problem, where each bag (*i.e.*, a WSI) has a label but each instance (*i.e.*, a patch) inside a bag has no label.

With the advent of convolutional neural networks (CNNs), numerous CNN-based MIL methods have been proposed for WSI diagnosis Chikontwe et al. (2020); Lerousseau et al. (2020); Xu et al. (2014); Feng & Zhou (2017); Ilse et al. (2018); Campanella et al. (2019); Lu et al. (2021); Li et al. (2021); Shao et al. (2021). These methods can be categorized into instance-level and embedding-level ones. Instance-level methods predict the pseudo-label of each instance based on bag-level labels, and then aggregate instance-level pseudo-labels to form the bag-level prediction Chikontwe et al. (2020). These methods usually have inferior performance due to their sensitivity to instance-level labels. Embedding-level methods convert each instance into a feature embedding, and then feed the feature embeddings from the same bag to an aggregator for bag-level prediction Xu et al. (2014); Feng & Zhou (2017); Ilse et al. (2018); Campanella et al. (2019); Lu et al. (2021); Li et al. (2021); Shao et al. (2021). Despite their notable success, these methods exhibit several major drawbacks. First, a WSI may present variable diagnostic information at different scales (Figure 1). For instance, a pathologist

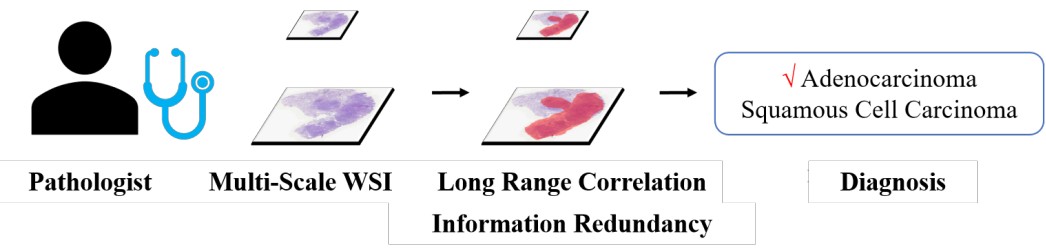

Figure 1: Reading behavior of pathologists.

may examine a WSI at multiple scales before making the final diagnosis, *e.g.*, determining if the tissue is necrotic in a global view and whether there is mitoses or microvascular proliferation in a local view **?**. Second, given the enormous size of each WSI, there inevitably exists long-range correlation among tissue/tumor regions that may be corrupted by partitioning the WSI into patches and extracting patch-level features independently. Third, for each WSI, only a small number of patches contain disease-positive regions, while the majority contain disease-negative regions, leading to severe information redundancy. Therefore, it is critical to select the most informative instances (patches) in each bag before aggregation.

To address these drawbacks, in this paper, we propose a state space model-based hierarchical multiple instance Learning (**Mamba-HMIL**) method for cancer diagnosis using WSI. Our **Mamba-HMIL** consists of three major parts. First, we deploy hierarchical encoders to extract multiscale features, mirroring the practice of a pathologist. Second, we employ a state space model (SSM) for feature aggregation to capture long-range correlations among tissue and tumor regions across thousands of patches while maintaining a manageable computational cost. Additionally, to ensure stable training, we incorporate a Mixture of Experts (MoE) and sequence fusion module to balance the contributions of each SSM sequence. Third, we insert an adaptive selection module to filter out disease-negative patches before classification. We verify the effectiveness of each component of our **Mamba-HMIL** and evaluate it against existing subtype classification and survival prediction methods using four public datasets. The contributions of this work are two-fold.

- We propose a novel solution to WSI classification, which extracts multiscale features, estimates the long-range correlation among tissue/tumor regions, and utilizes sparse selection to mitigate patch redundancy.
- The proposed **Mamba-HMIL** beats all competing methods on two public WSI classification datasets, setting the new state of the art.

## 2 RELATED WORK

Various MIL methods have been proposed to solve the weakly supervised classification task Zhou & Hua (2004). MIL was proposed for the first time and applied for drug activity prediction. Dietterich *et al.* Dietterich et al. (1997) compared three kinds of methods: a noise-tolerant algorithm, an "outside" algorithm, and an "inside-out" algorithm. The "inside-out" algorithm named region growing achieves the best results among these three methods. Maron *et al.* Maron & Ratan (1998) first used MIL in natural scene image classification. Zhou *et al.* Zhou et al. (2012) defined a multiple instance and multi-label (MIML) task for scene image classification. With the development of deep learning, a large number of deep learning based MIL is proposed to solve various tasks.

In particular, MIL has been widely used in digital pathological image analysis. With the development of deep learning, deep learning based MIL achieves great success in digital pathological image analysis Xu et al. (2014); Ilse et al. (2018); Campanella et al. (2019); Lu et al. (2021); Li et al. (2021); Shao et al. (2021). Xu *et al.* Xu et al. (2014) classified pathological images by establishing a deep MIL paradigm, where instance feature representations were operated by deep learning networks and aggregated by MIL. Pinheiro *et al.* Pinheiro & Collobert (2015) proposed a pooling-based method such as mean-pooling or max-pooling. Ilse *et al.* Ilse et al. (2018) proposed an attention-based deep MIL method, which was just a linear weighted combination. Campanella *et al.* Campanella et al.

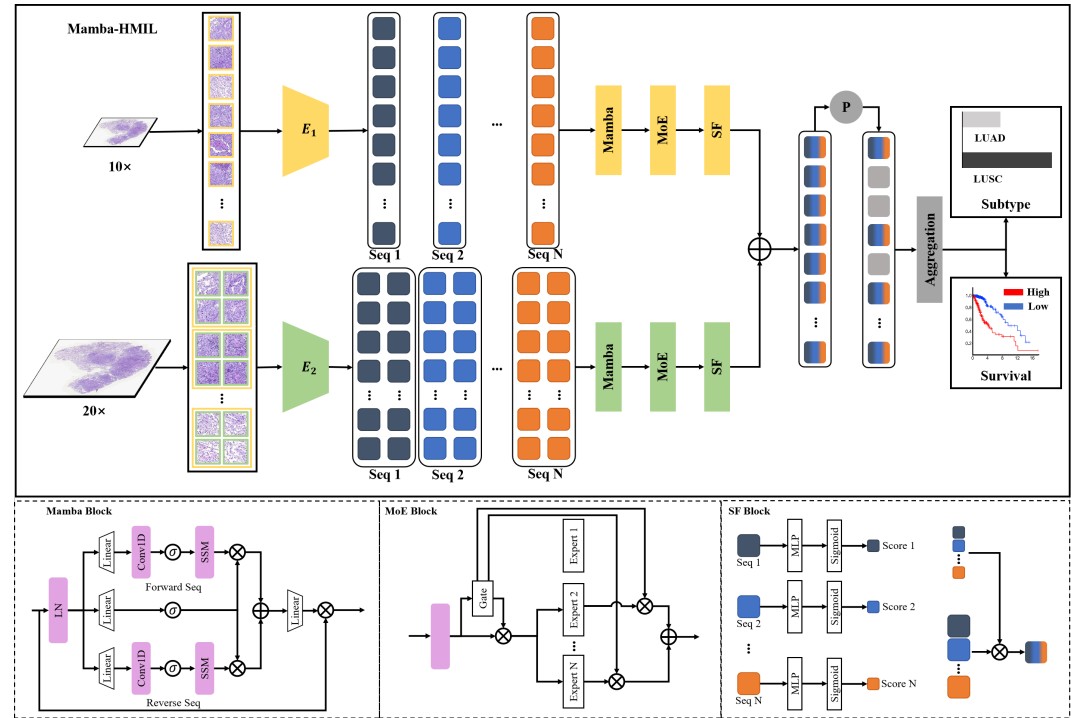

Figure 2: Framework of our proposed **Mamba-HMIL**, including three components: hierarchical feature extractor (HFE), state space model (Mamba), Mixture of Experts (MoE), sequence fusion (SF), and adaptive selection (AS) block. In particular, a WSI is first cropped into multiscale patches ($10\times$ and $20\times$), which are regarded as multiscale bags of instances. The level $10\times$ instances are passed through the feature extractor $E_1$ to produce level $10\times$ embeddings. By combining these embeddings in various ways, we generate different sequences Seq 1, Seq 2, ..., Seq N. These sequences are then fed into the Mamba, MoE, and SF blocks. These selected embeddings, together with the level $20\times$ embeddings, undergo hierarchical fusion processing to merge multiscale features. Subsequently, the fused sequence embeddings are filtered by the AS block, selecting those with a high probability of being positive. The embeddings with higher positive likelihood are retained and passed through the MLP head, culminating in a bag-level prediction.

(2019) proposed a recurrent neural network (RNN) based MIL aggregation that took the relation of neighboring instances into account. Li *et al*. Li et al. (2021) proposed a dual-stream MIL, which used the relation between the most possible positive instance and other instances, but ignored the correlation of other instances. Shao *et al*. Shao et al. (2021) developed a Transformer-based MIL that considered the correlation among instances, but its performance improvement is largely dependent on a pyramid convolutional block. Zhang*et al*. Zhang et al. (2022) proposed DTFD-MIL to use pseudo bags and feature distillation. Chen*et al*. Chen et al. (2022a) proposed a hierarchical self-supervised learning method for WSI classification. Yang*et al*. Yang et al. (2024) explored Mamba-MIL, and used Bi-Mamba for sequence correltaion.

## 3 METHOD

### 3.1 MULTIPLE INSTANCE LEARNING

MIL is an effective method to classify bags which contain uncertain number of instances. According to the hypothesis of MIL for binary classification task, each bag has a label. If a bag contains at least one positive instance, the label of bag is positive. On the other hand, if the instances in a bag are all negative, the label of bag is negative. Supposing that $X$ is a bag with label $Y \in \{0, 1\}$, which contains several instances $\{x_1, x_2, \cdots, x_n\}$ with labels $\{y_1, y_2, \cdots, y_n\}$, $y_i \in \{0, 1\}$, an MIL task

---

**Algorithm 1** SSM+SS processing flow.

---

**Input:** A bag of instance embeddings $H_{l-1} \in \mathbb{R}^{1 \times N \times D}$
1: State Space Model (SSM)
2: $H'_{l-1} \leftarrow \text{Norm}(H_{l-1})$
3: **for** $i$ in {Forward,Reverse} **do**
4:     $H_i \leftarrow \text{SSM}(\text{SiLU}(\text{Conv1D}(\text{Linear}(H'_{l-1}))))$
5: **end for**
6: $H_s \leftarrow \text{SiLU}(\text{Linear}(H'_{l-1}))$
7: $H_{Forward} \leftarrow H_{Forward} \bigotimes H_s$
8: $H_{Reverse} \leftarrow H_{Reverse} \bigotimes H_s$
9: $H_l \leftarrow \text{Linear}(H_{Forward} \bigoplus H_{Reverse}) + H_{l-1}$
**Output:** Instance embeddings $H_L \in \mathbb{R}^{N \times D}$

---

can be defined as

$$Y = \begin{cases} 1, & \text{if } \sum y_i = 0, \\ 0, & \text{otherwise.} \end{cases} \tag{1}$$

There are two main approaches in an MIL operator: the instance-level approach and the embedding-level approach. These two approaches share a similar expression. The bag probability is regarded as a score function $S(X)$, which is defined as

$$S(X) = g\left( \underset{x_i \in X}{\sigma} (f(x_i)) \right). \tag{2}$$

For instance-level approach, $f(\cdot)$ is an instance-level classifier that returns each instance score. $\sigma(\cdot)$ acts as a function to aggregate instance scores. $g(\cdot)$ is the identity function. For, embedding-level approach, $f(\cdot)$ maps instances to a low-dimensional embedding. $\sigma(\cdot)$ is used to obtain a bag representation that is independent of the number of instances. $g(\cdot)$ is a bag-level classifier.

### 3.2 HIERARCHICAL FEATURE EXTRACTOR

The feature extractor is flexible to various deep learning networks. In this paper, we choose ResNet-50 He et al. (2016) and Vision Transformer Dosovitskiy (2020) as the feature extractor for comparing with other methods easily.

**ResNet-50** consists of a $7 \times 7$ convolutional (Conv) layer, a $3 \times 3$ max pooling layer, four stages of residual blocks (each residual block is stacked by a fixed mode of $1 \times 1$, $3 \times 3$ and $1 \times 1$ Conv layers, and four stages contain 3, 4, 6 and 3 residual blocks respectively), a global average pooling layer, a fully connected layer (FC) and softmax. The FC layer and softmax are removed and the remaining part is used as the feature extractor. We choose ResNet-50 pre-trained on ImageNet as the basic model.

**ViT** consists of a linear projection layer followed by Transformer blocks, each containing a multi-head self-attention (MHSA) mechanism, a feed-forward network (FFN), and two layer normalization (LN) stages. Residual connections are applied after both the MHSA and FFN layers to improve gradient flow. We choose ViT-Large pre-trained by UNI Chen et al. (2023) as the feature extractors.

$E_1$ and $E_2$ are the same encoders, which are used to extract different scales of features ($10\times$ and $20\times$) of WSIs.

### 3.3 STATE SPACE MODEL

The state space model (SSM) Mamba Gu & Dao (2023) maps 1-dimensional function or sequence $x(t) \in \mathbb{R} \mapsto y(t) \in \mathbb{R}$ through a hidden state $h(t) \in \mathbb{R}^N$. SSM is represented as the linear ordinary differential equation (ODE):

$$x'(t) = \mathbf{A}h(t) + \mathbf{B}x(t), \tag{3}$$

$$y(t) = \mathbf{C}h(t), \tag{4}$$

where $\mathbf{A} \in \mathbb{R}^{N \times N}$ $\mathbf{B} \in \mathbb{R}^{N \times 1}$ and $\mathbf{C} \in \mathbb{R}^1$ are state parameters. The SSM consists of three branches: the forward sequence flow, the reverse sequence flow, and a nonlinear flow. The forward

Table 1: Performance comparison of subtype classification on TCGA-NSCLC and TCGA-RCC.

| Method | TCGA-NSCLC | | TCGA-RCC | |
|---|---|---|---|---|
| | ACC | AUC | ACC | AUC |
| MIL | 0.817±0.009 | 0.858±0.021 | 0.847±0.018 | 0.941±0.010 |
| ABMIL | 0.821±0.017 | 0.871±0.033 | 0.857±0.011 | 0.951±0.004 |
| Mamba+ABMIL | **0.836±0.019** | **0.905±0.027** | **0.896±0.019** | **0.955±0.008** |
| CLAM-MB | 0.853±0.012 | 0.933±0.007 | 0.897±0.010 | 0.979±0.008 |
| Mamba+CLAM-MB | **0.871±0.008** | **0.936±0.009** | **0.913±0.016** | **0.982±0.006** |
| DSMIL | 0.828±0.015 | 0.897±0.015 | 0.863±0.021 | 0.955±0.003 |
| Mamba+DSMIL | **0.846±0.017** | **0.918±0.009** | **0.901±0.017** | **0.974±0.007** |
| TransMIL | 0.813±0.013 | 0.881±0.020 | 0.890±0.014 | 0.962±0.009 |
| DTFD-MIL | 0.873±0.025 | 0.927±0.018 | 0.921±0.010 | 0.985±0.004 |
| Mamba-MIL | 0.863±0.014 | 0.924±0.011 | 0.913±0.009 | 0.974±0.009 |
| HIPT | 0.878±0.007 | 0.939±0.016 | 0.930±0.010 | 0.979±0.008 |
| HIGT | 0.872±0.011 | 0.925±0.019 | 0.919±0.010 | 0.974±0.007 |
| Mamba-HMIL | **0.884±0.025** | **0.944±0.012** | **0.936±0.011** | **0.989±0.008** |
| Mamba-HMIL+UNI | **0.911±0.008** | **0.964±0.008** | **0.946±0.004** | **0.989±0.001** |

Table 2: Performance comparison of survival prediction on TCGA-BRCA and TCGA-BLCA.

| Method | Modality | TCGA-BRCA | TCGA-BLCA |
|---|---|---|---|
| SNN | G | 0.565±0.035 | 0.517±0.053 |
| ABMIL | P | 0.593±0.047 | 0.584±0.068 |
| Mamba+ABMIL | P | **0.627±0.053** | **0.611±0.038** |
| CLAM-MB | P | 0.635±0.044 | 0.623±0.032 |
| Mamba+CLAM-MB | P | **0.657±0.047** | **0.633±0.061** |
| DSMIL | P | 0.607±0.033 | 0.601±0.029 |
| Mamba+DSMIL | P | **0.625±0.053** | **0.627±0.048** |
| Propoise | G+P | 0.644±0.035 | 0.634±0.052 |
| MCAT | G+P | 0.659±0.046 | 0.652±0.071 |
| CMTA | G+P | 0.684±0.042 | 0.661±0.054 |
| MOTCat | G+P | 0.663±0.045 | 0.657±0.058 |
| PIBD | G+P | 0.696±0.071 | 0.643±0.062 |
| Mamba-HMIL | P | 0.661±0.035 | 0.651±0.042 |
| Mamba-HMIL | G+P | 0.677±0.039 | 0.658±0.052 |
| Mamba-HMIL+UNI | P | 0.684±0.050 | 0.672±0.041 |
| Mamba-HMIL+UNI | G+P | **0.698±0.068** | **0.682±0.063** |

and reverse sequence flows are the same, which comprise a linear layer, a 1-dimensional convolution layer (Conv1D), a SiLU activation function, and the SSM layer. The nonlinear flow contains a linear layer and a SiLU activation function. The features from the forward/reverse flow and the nonlinear flow are merged by the Hadamard product. After that, the features are added together and transformed to the output embeddings by a linear layer. The workflow of Mamba is shown in Algorithm 1

Table 3: Ablation study for HFE block on TCGA-NSCLC and TCGA-RCC datasets.

| HFE level | TCGA-NSCLC | | TCGA-RCC | |
|---|---|---|---|---|
| | ACC | AUC | ACC | AUC |
| $5\times+10\times$ | 0.802 | 0.847 | 0.821 | 0.939 |
| $10\times+20\times$ | **0.829** | **0.895** | **0.855** | **0.944** |
| $5\times+10\times+20\times$ | 0.821 | 0.880 | 0.847 | 0.935 |

## 3.4 Mixture of Experts Module

For stable training, we use the Mixture of Experts (MoE) for multi-squence fusion. The gating mechanism is a simple linear layer, which computes relevance scores for each expert. The gating mechanism then activates the top-k experts based on these scores, directing the input through only those experts. The sequence passes through the selected experts, the outputs from these experts are combined. The aggregation is a weighted combination based on the gate's selection scores.

## 3.5 Adaptive selection Module

A adaptive selection (AS) module is used to discard redundant negative instances. It contains a MLP layer and a Sigmoid function. We utilize the AS module to compute a weight score for each sequence, and all sequences are then aggregated based on their respective weights. We set a temperature parameter P to balance the number of instances in each bag.

## 4 Experiments and Results

In this section, two publicly available clinical datasets in the cancer genome atlas (TCGA) **?** are used to demonstrate the effectiveness of our **Mamba-HMIL** in WSI classification. We also conduct an ablation study on thest two datasets.

## 4.1 Experiment Setup and Implementation Details

**Experiment setup and evaluation metrics.** In our experiment, each WSI of both two pathological image datasets is cropped into $256 \times 256$ non-overlapping patches to form bags with magnifications of $10\times$ and $20\times$, where the background region (entropy $< 5$) is discarded. Beyond that, we utilize two standard evaluation metrics to evaluate the classification performance, which are accuracy (ACC) and the area under the receiver operator characteristic curve (AUC).

**Implementation details.** Experiments are implemented on the device NVIDIA GTX 3080 GPU, Intel(R) Xeon(R) CPU E5-2690 v4 @ 2.60GHz, in Python 3.10 on Anaconda with CUDA 12.1 and Pytorch 2.1.0. We use Adam optimizer with learning rate 2e-4 to optimize SSM+SS training. The batch size is 1 and the maximum epoch is 200. In order to find the most suitable training parameters, cross-validation is formed from the whole slides in all the TCGA datasets.

## 4.2 Datasets.

**Subtype Classification.** TCGA lung Non-small-cell cancer dataset (TCGA-NSCLC) includes two sub-type projects, Lung Adenocarcinoma (TCGA-LUAD, 541 slides) and Lung Squamous Cell Carcinoma (TCGA-LUSC, 512 slides), with a total of 1,053 diagnostic WSIs available from the National Cancer Institute Data Portal. Each WSI is cropped into $256 \times 256$ non-overlapping patches at $5\times$, $10\times$, and $20\times$ magnification.

TCGA kidney chromophobe renal cell carcinoma cancer dataset (TCGA-RCC) consists of three kinds of tumors, kidney renal clear cell carcinoma (TCGA-KIRC, 519 slides), kidney renal papillary cell carcinoma (TCGA-KIRP, 300 slides) and kidney chromophobe renal cell carcinoma (TGCA-KICH, 121 slides). We use the same pre-processed operation of the TCGA-NSCLC dataset.

We follow the previous work and use 4-fold cross-validation to conduct our experiments. Both datasets are split into training, validation, and testing sets by the ratio of 6:1.5:2.5.

**Survival Prediction.** TCGA-BRCA (1022 cases) and TCGA-LUSC (373 cases) are used for the evaluation of survival prediction. 5-fold cross validation are used in our experiments.

## 4.3 RESULTS

We conducted a comparative evaluation of our proposed **Mamba-HMIL** against eight state-of-the-art methods, including ABMIL Ilse et al. (2018), CLAM-SB Lu et al. (2021), DSMIL Li et al. (2021), TransMIL Shao et al. (2021), DTFD-MIL Zhang et al. (2022), Mamba-MIL Yang et al. (2024), HIPT Chen et al. (2022a), and HIGT Guo et al. (2023). As outlined in Table 1, our **Mamba-HMIL** demonstrates superior performance, improving accuracy (ACC) by 0.9% and area under the ROC curve (AUC) by 0.5% on the TCGA-NSCLC dataset. Similarly, it improves ACC by 0.6% and AUC by 0.4% on the TCGA-RCC dataset. These improvements, although incremental, highlight the robustness of **Mamba-HMIL** in addressing the complexities of these datasets.

We also compare our method against five state-of-the-art survival prediction models, including Propoise Chen et al. (2022b), MCAT Chen et al. (2021), CMTA Zhou & Chen (2023), MOTCat Xu & Chen (2023), and PIBD Zhang et al. (2024). Our proposed model, **Mamba-HMIL**, is built upon the CLAM architecture. When compared to MIL-based methods that rely solely on pathological image data, **Mamba-HMIL**outperforms all other methods, demonstrating superior performance in survival prediction, as outlined in Table 2. Furthermore, when compared to multi-modality methods that incorporate genomic data, our model produces competitive results. Notably, when using pre-trained features, **Mamba-HMIL**achieves the highest C-Index scores on both the TCGA-BRCA and TCGA-BLCA datasets. This highlights the effectiveness of **Mamba-HMIL**in leveraging pre-trained features for improved survival prediction, making it a strong contender in both single-modality and multi-modality scenarios.

Additionally, we integrated the Mamba block into existing models such as ABMIL, CLAM-MB, and DSMIL, which led to general performance enhancements across both tasks. The inclusion of the Mamba block in these established models underscores its effectiveness in capturing more nuanced features and improving overall performance, making it a valuable addition to multiple architectures. This comparison not only validates the efficacy of **Mamba-HMIL** but also shows the potential of the Mamba block as a versatile component in other MIL frameworks.

## 4.4 ABLATION STUDY

**Effectiveness of HFE.** In our experiment, we use one fold of the dataset to determine the optimal number of blocks for our model. We then compare the performance of hierarchical feature extractors (HFE) with single-level feature extractors, as outlined in Table 3. The comparison is carried out on two datasets, demonstrating the superior performance of our proposed method. Specifically, on the TCGA-NSCLC dataset, the hierarchical feature extractor leveraging both $10\times$ and $20\times$ magnification levels improves accuracy (ACC) by 2.7% and the area under the ROC curve (AUC) by 4.8%, compared to the single-level feature extractor. On the TCGA-RCC dataset, the same hierarchical approach leads to an improvement of 3.4% in ACC and 0.5% in AUC. These results highlight the efficacy of using multi-scale features, showcasing that hierarchical feature extraction significantly enhances both classification accuracy and robustness in capturing nuanced patterns across different datasets.

**Effectiveness of Mamba Block.** For our baseline, we select ImageNet pre-trained ResNet-50 and the ABMIL model to evaluate the performance of our method. One of our key goals is to determine the optimal number of Mamba layers for the best performance. As presented in Table 4, the model incorporating two Mamba layers produces the best results on both the TCGA-NSCLC and TCGA-RCC datasets. Specifically, **Mamba-HMIL** with two Self-Supervised Masking (SSM) blocks achieves an accuracy (ACC) of 0.836 and an area under the ROC curve (AUC) of 0.905 on the TCGA-NSCLC dataset. On the TCGA-RCC dataset, the model achieves an ACC of 0.896 and an AUC of 0.955. These figures represent significant improvements over the baseline models: an increase of 1.5% in ACC and 3.4% in AUC for TCGA-NSCLC, and gains of 3.9% in ACC and 0.4%

Table 4: Ablation study for the number of Mamba blocks on TCGA-NSCLC and TCGA-RCC datasets. ABMIL is chosen for the baseline with 0 Mamba blocks.

| Mamba layers | TCGA-NSCLC | | TCGA-RCC | |
|---|---|---|---|---|
| | ACC | AUC | ACC | AUC |
| 0 | 0.821±0.017 | 0.871±0.033 | 0.857±0.011 | 0.951±0.004 |
| 2 | **0.836±0.019** | **0.905±0.027** | **0.896±0.019** | **0.955±0.008** |
| 4 | 0.833±0.014 | 0.879±0.016 | 0.895±0.006 | 0.947±0.009 |
| 6 | 0.825±0.036 | 0.885±0.031 | 0.888±0.014 | **0.955±0.006** |
| 8 | 0.822±0.028 | 0.893±0.021 | 0.891±0.011 | 0.948±0.018 |

Table 5: Ablation study for the number of experts in MoE block and the fusion strategy of SF blocks.

| MoE Type | Experts | SF Type | TCGA-RCC | |
|---|---|---|---|---|
| | | | ACC | AUC |
| MoE | 16 | - | 0.881±0.021 | 0.964±0.011 |
| STMoE | 16 | - | 0.889±0.027 | **0.975±0.005** |
| PEER | $512^2$ | - | **0.894±0.009** | 0.973±0.004 |
| Sinkhorn | 16 | - | 0.862±0.034 | 0.951±0.032 |
| STMoE | 4 | - | 0.771±0.133 | 0.814±0.190 |
| STMoE | 8 | - | 0.783±0.140 | 0.801±0.184 |
| STMoE | 16 | - | **0.889±0.027** | **0.975±0.005** |
| STMoE | 32 | - | 0.887±0.014 | 0.938±0.011 |
| STMoE | 64 | - | 0.850±0.037 | 0.886±0.048 |
| STMoE | 16 | Mean | 0.883±0.019 | 0.950±0.012 |
| STMoE | 16 | Max-Mean | 0.899±0.020 | **0.978±0.005** |
| STMoE | 16 | GAS | **0.914±0.016** | **0.978±0.006** |

in AUC for TCGA-RCC. These results highlight the effectiveness of adding two Mamba layers into the basic model.

**Effectiveness of MoE Blocks.** We conducted an evaluation of various MoE models using the base ABMIL architecture to assess the performance, as shown in Table 6. The models tested include the basic MoE Shazeer et al. (2017), STMoE Zoph et al. (2022), PEER He (2024), and Sinkhorn Anthony et al. (2024). Among these, STMoE achieved the highest AUC, scoring 0.975, while PEER delivered the best accuracy (ACC) at 0.894. Despite PEER's strong performance in terms of accuracy, it employs a significantly higher number of experts ($512^2$) compared to STMoE, which utilizes only 16 experts. Given the substantial increase in computational complexity and resource demand associated with PEER's larger number of experts, we selected STMoE for further experimentation in order to maintain a balance between performance and efficiency. In subsequent experiments, our results indicate that STMoE with 16 experts delivers the best performance. Specifically, the 16-expert configuration outperformed the 32-expert variant, with improvements of 0.2% in accuracy and 3.7% in AUC. This demonstrates that increasing the number of experts beyond a certain point can lead to diminishing returns, making 16 experts the ideal choice for maximizing performance while minimizing computational overhead in our subsequent experiments.

**Effectiveness of SF Blocks.** To explore the most effective method for sequence fusion, we evaluated three different SF blocks: Mean, Max-Mean, and GAS. Each of these blocks was assessed for its ability to integrate information across sequences and improve model performance. Among the three, GAS emerged as the best-performing block in terms of ACC, achieving a score of 0.914. This highlights the robustness of the GAS block in accurately capturing relationships within the sequence data. When comparing the AUC, both the Max-Mean and GAS blocks delivered identical top-tier results with an AUC of 0.978. However, there was a notable difference in the stability of these models, as reflected by the standard deviation. The Max-Mean block demonstrated a lower standard

Table 6: Ablation study for the token selection block on TCGA-RCC dataset. We choose CLAM with Top-K token selection as the baseline.

| AS Type | Value | TCGA-RCC | |
|---|---|---|---|
| | | ACC | AUC |
| Top-K | K=8 | 0.899±0.013 | 0.979±0.003 |
| Adaptive | P=0.7 | 0.905±0.026 | 0.980±0.004 |
| Adaptive | p=0.8 | **0.916±0.009** | **0.983±0.004** |
| Adaptive | p=0.9 | 0.895±0.020 | 0.975±0.009 |

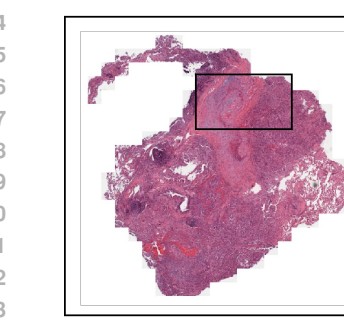
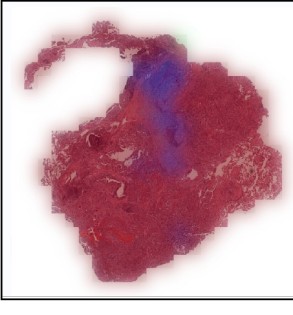
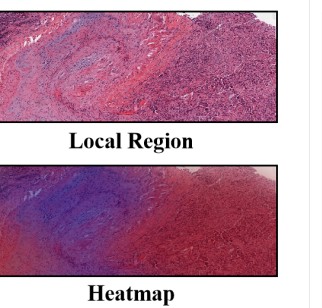
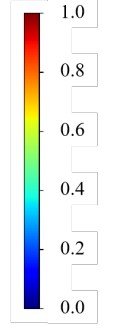

**WSI**  **Heatmap**

**Local Region**

**Heatmap**

Figure 3: The visualization of global WSI and local region by our **Mamba-HMIL**.

deviation compared to GAS, indicating more consistent performance across different experimental runs.

**Effectiveness of AS blocks.** In our study, we selected CLAM with Top-K selection as the baseline model due to its unique inclusion of a token selection block, which differentiates it from other models. However, the fixed token selection approach (K=8) does not account for the variability in the number of positive tokens present in different WSIs. Recognizing this limitation, we introduced an adaptive selection model (AS) that adjusts the number of selected tokens based on the characteristics of each WSI, rather than using a fixed value. We evaluated different values for the parameter P (0.7, 0.8, and 0.9), which controls the proportion of selected tokens. As shown in Table 5, we find that P=0.8 yielded the best results, with an ACC of 0.916 and an AUC of 0.983. These results represent a significant improvement over the baseline CLAM model, with a 1.7% increase in ACC and a 0.4% increase in AUC.

Together, these results highlight the importance of a multi-faceted approach in designing a model for pathological image analysis, combining hierarchical feature extraction, global correlation modeling, sequence weighting, and instance selection to achieve superior results.

## 5 CONCLUSION

In this paper, we have proposed **Mamba-HMIL** to solve the WSI classification task. **Mamba-HMIL** consists of three stages: the hierarchical feature extractor, the state space model, and the sparse selection block. We design the hierarchical feature extractor to obtain multi-scale features like a pathologist. The state space model is then utilized to calculate the correlation among instances, and the sparse selection module is used to select the instance embeddings with high positive probability and aggregate for a WSI-level prediction. Extensive experiments have been performed on two WSI classification datasets. The experimental results indicate that **Mamba-HMIL** can dramatically improve the performance of WSI-level classification. Our future work will focus on prognostic analysis and validation of other external data.

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
