# OpenReview forum: "Mamba-HMIL: Hierarchical Multiple Instance Learning via State Space Model for Whole Slide Image Diagnosis"
_ICLR.cc/2025/Conference — ICLR 2025 Conference Withdrawn Submission_

### Official Review · Reviewer_xc6o · 2024-11-03

**Soundness:** 2
**Presentation:** 2
**Contribution:** 2
**Rating:** 3
**Confidence:** 3

**Summary:**

The paper proposes MambaHMIL, a variant of MambaMIL which uses Hierarchical feature extraction, Mixture of Experts and adaptive fusion of sequences to improve performance over existing MIL baselines.

**Strengths:**

The paper discusses use of State space models for MIL with applications in pathology. While there is some prior work - MambaMIL which discusses state space models for MIL, the authors expand on it by incorporating multi-resolution feature aggregation, MoE and adaptive selection of sequences.
The authors compare against several existing MIL models to evaluate the approach across two common WSI level benchmark problems - subtyping and survival prediction.

**Weaknesses:**

The paper doesn't provide good motivation on why the specific additions/design choices made are relevant in the context of the problems and why they help. It primarily feels the authors did extensive hyper-parameer selection on the datasets and its unclear how these parameters or choices generalize. There isn't much discussion into why such parameters could be optimal for the dataset or the problem, which makes it challenging to come away with clear take-aways.
For example how does Mamba help with with performance and how does it compare with Attention based aggregation. The authors do show some comparison against HIPT, TransMIL here but its hard to compare and put these in context without discussing #params.
Similarly there are some ablations adding mamba to existing MIL models but there isnt much discussion on how its helping and if the improvements are just due to additional params
How multi-resolution helps and why adding more resolutions hampers performance?
What are the different sequences and what is the relevance of their fusion/aggregations.

It also doesn't give clear details on how some these these are implemented. Added some of these in questions section below.
The authors mention MoE was added to stabilize training, but its unclear what the instability was and how it improved stability.
Its also unclear how much MoE helped as there are no ablations with/out MoE.

The details around SSM, mixture of experts and adaptive selection are also not described clearly with no clear equations to describe the formulation.

**Questions:**

- For Hierarchical feature extractor
  - How are the features from different resolutions aggregated? Is it addition or concatenation.
  - How do the number of parameters change with the number of resolutions? Is the number of model parameters controlled for when comparing performance?
Given this is one of the key contributions, the lack of discussion or detail around this makes it hard to put results in context.

- How are the different Sequences of instances (s, s2, .. sn) generated for input to Mamba.

- How is the visualization in Figure 3 generated? There is no reference to the figure anywhere in the paper.

- Its unclear what GAS aggregation is? There is no reference or explanation about it.

---

### Official Review · Reviewer_XdkQ · 2024-11-04

**Soundness:** 1
**Presentation:** 2
**Contribution:** 1
**Rating:** 3
**Confidence:** 5

**Summary:**

The authors present a state-space model-based hierarchical multiple instance learning (Mamba-HMIL) approach for cancer diagnosis using whole slide images (WSI), which involves three stages. In the first stage, hierarchical encoders are used to extract multiscale features. In the second stage, a state-space model (SSM) aggregates features to assess correlations among instances. The third stage introduces an adaptive selection module that filters out disease-negative patches prior to classification. The proposed method was evaluated on four public datasets for subtype classification and survival prediction, where it was benchmarked against existing approaches.

**Strengths:**

No notable strengths were identified in this work.

**Weaknesses:**

1. **Lack of novelty**: This work appears to be a straightforward combination of existing methods. The hierarchical encoder is similar to DSMIL [1], and the Mamba architecture and mixture of experts (MoE) module are identical to previously established designs. The adaptive selection (AS) module consists only of an MLP layer and a Sigmoid function. Additionally, this is not the first application of Mamba for WSI analysis. Overall, this approach lacks substantial innovation.

2. **Insufficient comparison with existing SSM methods**: The paper does not provide a thorough comparison with similar state-space model-based approaches, such as Vim4Path [2] and MamMIL [3].

3. **Writing quality**: The manuscript appears to lack careful proofreading. For example, there are confusing question marks on Lines 67 and 297.

[1] Li, Bin, Yin Li, and Kevin W. Eliceiri. "Dual-stream multiple instance learning network for whole slide image classification with self-supervised contrastive learning." In *Proceedings of the IEEE/CVF conference on computer vision and pattern recognition*, pp. 14318-14328. 2021.

[2] Nasiri-Sarvi, Ali, Vincent Quoc-Huy Trinh, Hassan Rivaz, and Mahdi S. Hosseini. "Vim4Path: Self-Supervised Vision Mamba for Histopathology Images." In *Proceedings of the IEEE/CVF Conference on Computer Vision and Pattern Recognition*, pp. 6894-6903. 2024.

[3] Fang, Zijie, Yifeng Wang, Ye Zhang, Zhi Wang, Jian Zhang, Xiangyang Ji, and Yongbing Zhang. "Mammil: Multiple instance learning for whole slide images with state space models." *arXiv preprint arXiv:2403.05160* (2024).

**Questions:**

1. Can the authors clarify the motivation behind this work, given that multiple studies have already applied Mamba to WSI analysis?

2. The performance of Mamba-MIL and HIPT on NSCLC is inconsistent with the results reported in the original papers, where both achieved an AUC above 0.95. I did not verify all baseline methods in this paper, but the authors should thoroughly review the experimental results and explain why the baseline methods underperformed significantly compared to the original studies.

3. In Section 4.4, the authors compare Mamba-HMIL to ABMIL with an ImageNet-pretrained encoder to evaluate the effectiveness of the Mamba Block. Is this a fair comparison?

---

### Official Review · Reviewer_b2nH · 2024-11-08

**Soundness:** 1
**Presentation:** 1
**Contribution:** 1
**Rating:** 1
**Confidence:** 5

**Summary:**

This paper presents hierarchical multiple instance learning using state space model for whole slide image diagnosis. The method propose to use several components such as hierarchical feature extractor, the state space model, and mixture of experts. The experiments are performed on two datasets.

**Strengths:**

This paper addresses an important problem of  WSI diagnosis

**Weaknesses:**

This paper is poorly written, has no contribution and is just the combination of different components without any clear motivation and reasoning. I would request the authors to clearly explain the reason behind choosing each component of approach and re-write the paper for better clarity.

**Questions:**

No questions. The paper  needs to be improved significantly.

---

### Official Review · Reviewer_UNQv · 2024-11-08

**Soundness:** 3
**Presentation:** 3
**Contribution:** 2
**Rating:** 6
**Confidence:** 5

**Summary:**

This work presents Mamba-HMIL, a hierarchical MIL method leveraging the state space modeling for weakly-supervised tasks in computational pathology. MAMBA-HMIL includes several components: (1) state-space modeling (Mamba), (2) Mixture of Experts (MoE) blocks, and (3) sequence fusion / adaptive fusion blocks. Mamba-HMIL is evaluated on cancer subtyping and survival prediction tasks, and compared with relevant baselines in the literature (ABMIL, CLAM, DSMIL, HIPT, and Mamba extensions to MIL). Ablation experiments for each component is performed.

**Strengths:**

- Overall, the experimental design is comprehensive and thorough. Mamba-HMIL is compared against many-to-all relevant baselines in the literature, from simple permutation-invariant pooling baselines (ABMIL, CLAM-MB), to Transformer MIL architectures that learn token dependencies (TransMIL, HIPT), to also direct extensions of Mamba applied to MIL architectures (Mamba+ABMIL, Mamba+DSMIL, Mamba+CLAM-MB) as well as Mamba-MIL. Evaluation on survival prediction is also appreciated and validates the strength of Mamba-HMIL in learning context-aware features for understanding the tumor microenvironment. Good attention-to-detail to the survival prediction baselines in evaluating recent SOTA early multimodal fusion architectures like MCAT and PIBD. This study also presents good depth of experiments, in not only ablating the components of Mamba-HMIL (MoE, SF/AF), but also validating other components in MIL including different pretrained encoders (ResNet-50 vs UNI) and hierarchical feature extraction (10X, 20X, 10X+20X).
- While direct extensions of Mamba are not a technical novelty, Mamba-HMIL has good performance gains and consistently achieves the best performance across all tasks. These performance gains are on top of comparisons to MIL architectures with Mamba extensions, which suggests that the architecture modifications are not ad hoc.
- Interestingly, unimodal Mamba-HMIL outperforms many multimodal fusion comparisons, including Mamba+CLAM-SB, PORPOISE, and MCAT. This is a good finding that should be highlighted more.

**Weaknesses:**

- Though method seems strong, many of the components of Mamba-HMIL itself are either not novel or not studied enough to demonstrate why we see strong improvement in performance. I would like to understand how these components "stabilize SSM training". It is not clear how SSM training is unstable when direct extension of Mamba works quite well for all MIL architectures across all tasks.
- I cannot review the code for this submission. As many of the contributions are empirical, it would be valuable to validate the contributions of this work empirically before acceptance.
- Is it possible to visualize token-to-token interactions by Mamba-HMIL, besides attention weights from global attention pooling? How do token interactions change across different Mamba layers?
- There are several outstanding typos in this work. There is missing-or-extra spacing after Mamba-HMIL on many lines. There are many misspellings like "Propoise". In the data description of survival prediction on Line 327, TCGA-LUSC instead of TCGA-BLCA is written. Many citations are missing and marked as ?.

**Questions:**

See above.

---

### Note · Authors · 2024-11-15

I have read and agree with the venue's withdrawal policy on behalf of myself and my co-authors.